# Plasmid-borne *mcr-1* and replicative transposition of episomal and chromosomal *bla*$_{NDM-1}$, *bla*$_{OXA-69}$, and *bla*$_{OXA-23}$ carbapenemases in a clinical *Acinetobacter baumannii* isolate

Masego Mmatli,[1] Nontombi Marylucy Mbelle,[1,2] John Osei Sekyere[1,3]

**ABSTRACT**  A multidrug-resistant clinical *Acinetobacter baumannii* isolate with resistance to most antibiotics was isolated from a patient at an intensive care unit. The genetic environment, transcriptome, mobile, and resistome were characterized. The MicroScan system, disc diffusion, and broth microdilution were used to determine the resistance profile of the isolate. A multiplex PCR assay was also used to screen for carbapenemases and *mcr*-1 to -5 resistance genes. Efflux-pump inhibitors were used to evaluate efflux activity. The resistome, mobilome, epigenome, and transcriptome were characterized. There was phenotypic resistance to 22 of the 25 antibiotics tested, intermediate resistance to levofloxacin and nalidixic acid, and susceptibility to tigecycline, which corresponded to the 27 resistance genes found in the genome, most of which occurred in multiple copies through replicative transposition. A plasmid-borne (pR-B2.MM_C3) *mcr*-1 and chromosomal *bla*$_{PER-7}$, *bla*$_{OXA-69}$, *bla*$_{OXA-23}$ (three copies), *bla*$_{ADC-25}$, *bla*$_{TEM-1B}$, and *bla*$_{NDM-1}$ were found within composite transposons, ISs, and/or class 1 and 2 integrons on genomic islands. Types I and II methylases and restriction endonucleases were in close synteny to these resistance genes within the genomic islands; chromosomal genomic islands aligned with known plasmids. There was a closer evolutionary relationship between the strain and global strains but not local or regional strains; the resistomes also differed. Significantly expressed/repressed genes (6.2%) included resistance genes, hypothetical proteins, mobile elements, methyltransferases, transcription factors, and membrane and efflux proteins. The genomic evolution observed in this strain explains its adaptability and pandrug resistance and shows its genomic plasticity on exposure to antibiotics.

**IMPORTANCE**  A pandrug-resistant pathogen that was susceptible to only a single antibiotic, tigecycline, was isolated from a middle-aged patient in an ICU. This pathogen contained two plasmids and had a chromosome that contained portions that were integrated externally from plasmids. These genomic islands were rich with resistance genes, mobile genetic elements, and restriction-modification systems that protected the pathogen and facilitated gene regulation. The strain contained 35 resistance genes and 12 virulence genes. The strain was of closer evolutionary distance to several international strains suggesting that it was imported into South Africa. However, its resistome was unique, suggesting an independent evolution on exposure to antibiotic therapy mediated by epigenomic factors and MGE transposition events. The varied mechanisms available to this strain to overcome antibiotic resistance and spread to other areas and/or transfer its resistance determinants are worrying. This is ultimately a risk to public health, evincing the need for antibiotic stewardship.

**KEYWORDS**  colistin resistance, carbapenem, carbapenemase, last-resort antibiotics, non-fermenters, multi-drug resistance, RNAseq

**Peer Reviewers** Olga E. Khokhlova, State Research Center for Applied Microbiology and Biotechnology, Obolensk, Russia; Yingshun Zhou, Southwest Medical University, Luzhou, China

Address correspondence to John Osei Sekyere, j.oseisekyere@up.ac.za.

Nontombi Marylucy Mbelle passed away during the preparation of the manuscript.

The authors declare no conflict of interest.

See the funding table on p. 16.

*This paper is dedicated to the memory of Professor Nontombi Marylucy Mbelle (1960–2021).*

$A$ *cinetobacter baumannii* is an aerobic, coccobacillary rod, non-fermenting Gram-negative pathogen [1, 2]. It is an important, ubiquitous, and opportunistic pathogen found in both moist and dry conditions and is well distributed within nature, the nosocomial environment, and the human mucosal microbiome [3]. *A. baumannii* causes both community- and healthcare-associated infections (HAIs) [4, 5] including urinary tract, bloodstream, skin, and tissue, as well as ventilator-associated infections [6, 7]. These infections are usually difficult to treat and are fatal [8, 9] as *A. baumannii* can survive for prolonged periods in the hospital environment, facilitating its nosocomial spread [10]. This is achieved through either direct contact with an infected patient (patient-to-patient), contact with the hands of healthcare personnel, or indirectly by touching contaminated environmental surfaces [1, 3, 11]. Individuals at risk for *A. baumannii*-related HAIs are typically those who are immuno-deficient [7, 10], or are undergoing invasive procedures such as the use of mechanical ventilators, central venous or urinary catheters, as seen in the invasive care unit (ICU) [4, 10].

Carbapenem resistance in *A. baumannii* is mainly acquired through the production of oxacillinase-type carbapenemases, with $bla_{OXA-23}$-like and $bla_{OXA-48}$-like being the most prevalent β-lactamase (carbapenemase) [12, 13]. Hence, carbapenem-resistant *A. baumannii* (CRAB) [13] is mostly treated using colistin and tigecycline as the last option. While phosphoethanolamine (PEtN)-mediated CRAB has been reported in South Africa [14, 15], mobile colistin resistance (*mcr*) genes have been reported in *A. baumannii* in China [16], Brazil [17], Italy [18], Pakistan [7], Turkey [5], Europe, and Iraq [19]. Nevertheless, there have been no reports of *mcr*-producing *A. baumannii* in South Africa. Hence, this study presents the first report of an *mcr*-positive CRAB isolate (with multiple carbapenemases) from South Africa and to our knowledge, the first globally, using genomics, transcriptomics, epigenomics, and advanced bioinformatics to characterize its mechanisms of resistance and genome structure.

## MATERIALS AND METHODS

### Sample source and phenotypic resistance

A 53-year-old female patient at aICU of the Steve Biko Academic Hospital, a tertiary and quaternary hospital in Pretoria, South Africa, presented with a difficult-to-treat infection in 2017. A fluid aspirate from the patient was cultured on blood agar media for 24 hours at 37°C and subsequently identified in the laboratory; the isolate was labeled as R-B2.MM [20]. The Microscan Walkaway identification/antibiotic susceptibility testing system (Beckman Coulter Diagnostics, United States) with Panel Combo 68 was used to identify the species and antibiotic resistance profiles of the isolate. Carbapenem and colistin resistance of the isolate were confirmed using the disc diffusion (10 µg discs of ertapenem, meropenem, and imipenem) and broth microdilution (BMD) methods, respectively. The Clinical Laboratory Standards Institute (CLSI) [21] and the European Committee on Antimicrobial Susceptibility Testing (EUCAST) [22] breakpoints were used, respectively, for the non-colistin antibiotics and colistin BMD. The BMD assay was performed using colistin sulfate powder, according to the CLSI standards.

Ertapenem sulfate salt and colistin sulfate salt (Glentham Life Sciences, United Kingdom) were used for the BMD assay [19]. *Escherichia coli* ATCC 25922 and *Pseudomonas aeruginosa* ATCC 27853 were included as quality control strains. Both antibiotics were dissolved in sterile deionized water according to the manufacturers' instructions. The antibiotic concentrations tested were as follows: 128 µg/mL, 64 µg/mL, 32 µg/mL, 16 µg/mL, 8 µg/mL, 4 µg/mL, 2 µg/mL, 1 µg/mL, 0.5 µg/mL, and 0.25 µg/mL.

The BMD assay was performed in untreated 96-well polystyrene microtiter plates, with each well containing 100 µL of antibiotic dilution and Mueller-Hinton broth (MHB) or cation-adjusted MHB for ertapenem and colistin, respectively. Subsequently, a 0.5 MacFarland suspension of bacterial culture was prepared, diluted to 1:20 with sterile saline, and 0.01 mL was inoculated into each well. The plates also included sensitive and negative control wells.

## Efflux-pump inhibitors

The role of efflux pumps in the resistance mechanisms of the isolate was investigated using the BMD method and the following efflux-pump inhibitors (EPIs): verapamil, phenylalanine-arginine β-naphthylamide (PAβN), carbonyl cyanide 3-chlorophenylhydrazone (CCCP), reserpine, and ethylenediaminetetraacetic acid (EDTA). The change in carbapenem and colistin resistance minimum inhibitory concentrations (MICSs) in the presence of the EPIs were calculated. The BMD and agar plates were incubated at 37°C for 16–18 hours, and the minimum inhibitory concentration (MIC) was determined as the lowest antibiotic concentration without visible bacterial growth; the inhibition zones (for the disc diffusion tests) were used to determine carbapenem resistance using the CLSI breakpoints (21). The final concentrations of the antibiotic substrates in the broth were 1.5 µg/mL for CCCP, 4 µg/mL for VER, 25 µg/mL for PAβN, 20 µg/mL for RES, and 20 mM (pH 8.0) for EDTA. A ≥2-fold reduction in ertapenem and colistin MICs after EPI applications was indicative of significant efflux pump, metallo β-lactamase, and MCR activity and role in carbapenem or colistin resistance.

## Molecular characterization

Genomic DNA and RNA were, respectively, extracted from a 24 hour culture using Quick-DNA-fungal/bacterial MiniPrep kit (ZymoResearch) and Quick-RNA-fungal/bacterial MiniPrep kit (Zymo Research) according to the manufacturer's protocols. Prior to RNA extraction, the bacterial suspension was grown in a broth containing 0.5 mg/mL of ertapenem and 2 mg/mL of colistin for at least 12 hours. The RNA was converted into cDNA using Qiagen's cDNA synthesis kit.

Aliquots of the gDNA were used in a multiplex PCR screening test to identify the presence of *mcr* and carbapenemases using the primers in (Data set S1) and conditions already described in another study (20). The gDNA was sequenced using PacBio SMRT sequencing at 100× coverage while the cDNA was sequenced using Illumina MiSeq at a commercial sequencing facility. The PacBio reads (in fastQ format) were assembled using PacBio's hierarchical genome-assembly process (HGAP) software to obtain a fastA file, which was annotated with NCBIs Prokaryotic Genome Annotation Pipeline (PGAP). The methylation files (motifs and base modifications) of the genome were also obtained from PacBio's SMRTAnalysis MotifMaker software.

The species, MLST profile, resistome, mobilome, and epigenome, of the isolate were determined using NCBIs Average Nucleotide Identity (ANI) (23), Center for Genomic Epidemiology's MLST 2.0 (24), ResFinder 4.0 (25), PlasmidFinder 2.1 (26), MGE (mobile genetic element) (27), and Restriction-ModificationFinder 1.1 (28). The GenBank annotation files were downloaded and parsed through SnapGene Version 7.2.1 to illustrate the genetic environment of the resistance genes.

Genome map analysis was undertaken using the genome fastA file on the Proksee platform. The GC content, resistance genes, CRISPR/Cas systems, mobile genetic elements and horizontal gene transfer events, phageome, and genes/ORFs were annotated and superimposed on the map (29).

## Phylogenomics

*A. baumannii* strains that were classified as resistant by computational means or through laboratory analyses were selected from the PATRIC database. Those from Africa were separated and downloaded and those from other continents were also grouped: a selection of 199 strains were randomly obtained from the non-African group and the genomes from the African strains were downloaded. Finally, sections of the strain's genome were BLASTed to identify other strains that closely aligned with it. The genomes of these closely aligned strains through BLAST analysis were also downloaded for additional phylogenetic analysis. The downloaded genomes, together with this study's genome, were aligned using ClustalW and ≥1,000 coding sequences were used to phylogenetically analyze their evolutionary relationship using the randomized

axelerated maximum likelihood (RAxML) tool. Default parameters were used except that 1,000 genes were set as the minimum for all genomes and a bootstrap of 1,000 was used. A bootstrap value of ≥50% was defined as statistically significant. The Newick file was annotated using FigTree v1.4.4.

## Epigenomics

The Restriction Enzyme Database (REBASE) (28), hosted by the Centre for Epidemiology, was used to identify the restriction-modification system (RMS), which includes DNA methylation, restriction endonucleases, and their motifs (30). The methylation modifications and motifs were also determined using PacBio's MotifMaker software (28). PGAP annotations of the contigs also identified the restriction endonucleases (REs), methylases or methyltransferases (MTAses), and associated methylation genes in each contig. These annotations were visualized using SnapGene 7.2.1.

## RNA-sequencing data analysis

HTSeq-DeSeq2 was used to align, assemble, and evaluate the differential gene expression of the isolate. *A. baumannii* ATCC 1909 strain was used as the reference genome. The function of each gene was evaluated using the genome annotations of the reference strain on the PATRIC database. The statistical significance of the logarithmic (log2) fold change in expression or repression levels of the various coding sequences was measured by DESeq2 as q-values and *P*-values, with *P* values < 0.05 being defined as significant.

## RESULTS

### Identification, typing, and resistance profile

The isolate was identified by the MicroScan Walkaway identification/antibiotic susceptibility testing system (Beckman Coulter Diagnostics, United States) using Panel Combo 68 as a non-fermenting, MDR, and ESβL-producing isolate. The species was confirmed by NCBI's ANI (23) to be *A. baumannii*, with resistance to 22 out of the 25 antibiotics tested: amikacin, amoxicillin-clavulanate, ampicillin/sulbactam, ampicillin, aztreonam, cefepime, cefotaxime, cefoxitin, ceftazidime, cefuroxime, cephalothin, ciprofloxacin, colistin, ertapenem, fosfomycin, gentamicin, imipenem, meropenem, norfloxacin, nitrofurantoin, tobramycin, and trimethoprim-sulfamethoxazole. It was however sensitive to tigecycline and piperacillin-tazobactam had intermediate susceptibility to levofloxacin and nalidixic acid. The BMD and disc diffusion tests confirmed isolate R-B2.MM is resistant to both colistin (>128 µg/mL) and the carbapenems (zone diameter > 19 mm).

The multiplex PCR screening of the isolate detected *mcr*-1 and a *bla*$_{OXA-like}$ gene, while the whole-genome sequencing identified 27 resistance genes (or 31 resistance genes with variants): *aadA1* (six copies: three copies on chromosomal contig 1 and three copies on chromosomal contig 2), *aac(3)-Ia, aph(3)-Ia, aph(3")-Ib, aph(6)-Id* (two copies on the chromosome; contig 1), *armA, arr-2, dfrA1* (four copies: two copies on chromosomal contig 1 and two on chromosomal contig 2), *dfrA15, mcr-1.1* (two copies on plasmid contig 3), *mphE, msrE, qacE, qnrS1, sitABCD, strA, strB, sul1* (three copies on chromosomal contig 1), *sul2* (three copies on chromosomal contig 1), *sul3, cmlA1* (two copies: one on contig 1 and another on plasmid contig 3), *qnrS1, tet*(B), *tet*(M), *tet*(A) (two copies on plasmid contig 3), *bla*$_{PER-7}$, *bla*$_{OXA-69}$, *bla*$_{OXA-23}$ (three copies on chromosomal contig 1), *bla*$_{ADC-25}$, *bla*$_{TEM-1B}$, and *bla*$_{NDM-1}$. The resistance phenotype corresponded to the resistome in that the resistance genes identified correspond to the resistance profiles observed from the MicroScan and BMD (Data set S1).

The strain was found to belong to multilocus sequence types (MLSTs) ST1604, ST231, or ST1: ST1604 and ST231 from the Oxford MLST scheme and ST1 according to the Pasteur MLST Scheme. This strain contained 12 virulence genes: *cma, ompT, traT, iutA, iucC, iss, hlyF, iroN, sitA, cib, cvaC,* and *ipfA*.

Among the four EPIs, EDTA (71.3-fold change) and CCCP (2.5-fold change) had a significant MIC reduction (≥2-fold) on colistin while the rest did not. Notably, the *P.*

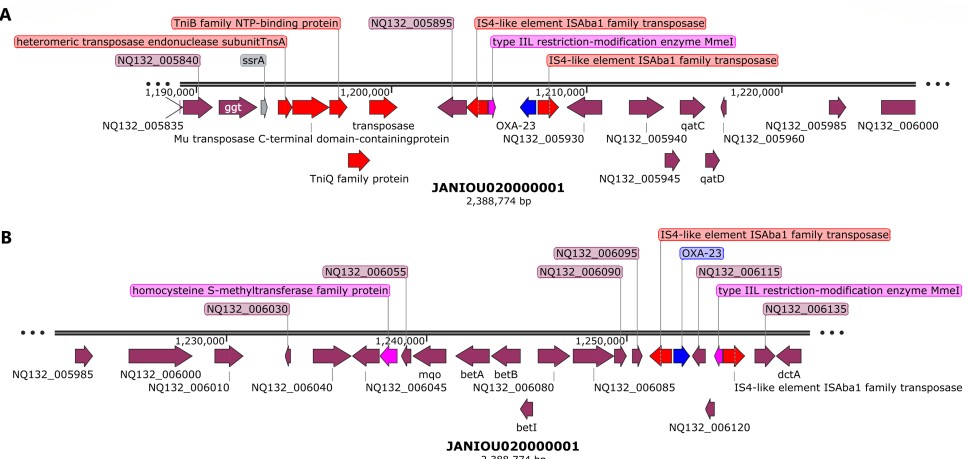

**FIG 1** Genetic environment of *blaOXA-23* carbapenemase on chromosomal contig 1 (1.19–1.22 Mb, 1.23–1.28 Mb). Mobile genetic elements (shown as red arrows) and methylases/restriction modification endonuclease (purple arrows) bracketing the *bla*$_{OXA-23}$ gene show that the gene is within a composite transposon. The same chromosomal contig has two *bla*$_{OXA-23}$ genes shown in panels A and B, with the genetic environment in panel A being different from that in panel B. In both cases, *bla*$_{OXA-23}$ was bracketed by IS*Aba1* and transposases.

*aeruginosa* ATCC 27853 had a significant reduction in colistin MIC in the presence of EDTA (twofold reduction) and reserpine (4.7-fold reduction). Contrarily, only EDTA caused a reduction (22.7-fold) in the MIC of ertapenem while PAβN and CCCP significantly reduced the MICs of ertapenem by twofold in the *E. coli* ATCC 25922.

## Genetic environment of resistance genes

Contigs 1 and 2 were chromosomal, with contig 1 containing more resistance genes than contig 2. Notably, contigs 1 and 2 also had their resistance genes clustered together in one region flanked by integron cassettes and composite transposons, forming a genomic island. On contig 1, the resistance genes clustered between 0 and 80 kb within integrases, recombinases, insertion sequences, transposons, DNA (cytosine) methyltransferases, and methylases (Fig. S1 through S5). Hence, all the resistance genes within this ~80 kb genomic island were bracketed by composite transposons that also included class 1 and class 2 gene cassettes. Within the genomic island on contig 1, *aadA, sat2*, and *dfrA1* were contiguous to each other and bracketed by Tns*D* and IS256 transposases. *aph(6')-I* was flanked by a class 2 integrase (*IntI2*) and IS*Vsa3* transposase, followed in close synteny by *sul2*, N-6 DNA methylase, IS*26*, and class 1 integrase (*IntI1*) in the reverse orientation to the *IntI*2. Following this *IntI*1 integron cassette were batteries of *IS*s, transposases, and resistance genes such as *arr-2, cmlA5, QacE, sul1, bla*$_{PER}$*, sul1, armA, msr(E), mph(E), tetR(B)*, and *sul2*. Instructively, there were two copies of *sul1* and *sul2* genes within the genomic island (Fig. S2 and S3).

Indeed, the whole-genomic island of ~80 kb seems to be a composite transposon and an episome as a BLAST analysis showed that it aligned with several plasmids such as Ab04-mff plasmid pAB04-1 (CP012007.1) and chromosomes such as that of *A. baumannii* AR_0083 (CP027528.1). Moreover, replicative transposition was observed at 1.19–1.22 Mb, 1.23–1.28 Mb, and 1.32–1.33 Mb regions on contig 1 (Fig. 1 and 2A) with *bla*$_{OXA-23}$ flanked by a type IIL restriction-modification enzyme *MmeI* and two IS*Aba1* transposases at both sides, forming a composite transposon. In Fig. 1A, additional Tn*SA*, Tn*iB*, Tn*iQ,* and Mu transposases were in close proximity to the composite IS*Aba1*-flanked transposases while a homocysteine S-methyltransferase was found in close proximity to the same composite transposon in Fig. 1B. Thus, three copies of *bla*$_{OXA-23}$ within an IS*Aba1*-flanked composite transposon were present in the genome owing to this replicative transposition. These three copies ranged from 1.19 to 1.34 Mb (Fig. 1 and

2A), which aligned to both chromosomes (including that of *A. baumannii* AR_0083 [CP027528.1]) and an unnamed plasmid (90% query cover and 97.7% nucleotide identity) from *A. baumannii* 2021CK-01407 (CP104448.1) when BLASTed.

Another genomic island with a composite transposon flanking *aph(3″)-Ib, aph(6)-Id*, *bla*$_{NDM-1}$:*ble*, and *sul2* was found between 328 and 353 kb on contig 1, which aligned to both *A. baumannii* plasmids and chromosomes: CP027528.1, AP031576.1, CP130627.1, CP035935.1, and CP090865.1. The composite transposon comprised of IS*1006*, recombinase, IS*Aba1*, IS*30*, IS*91*, and IS*Vsa3* with IS*30:bla*$_{NDM-1}$:*ble* synteny being present (Fig. 2B). A third genomic island was also identified on contig 1 between 455 kb and 500 kb (Fig. 3A), which comprised of a class 1 integron (*IntI1*), a recombinase, DNA adenine methylase, IS*Ppu12* and IS*26*-IS*6* transposases, and *aph(3″-Ia*, GNAT N-acetyltransferase (two copies), *aadA1, QacEΔ1, aac(3″)-Ia, aadA1*, and *sul1* resistance genes. This genomic island was in close synteny with an upstream *parC* gene and mostly aligned with chromosomes and a single plasmid (from *A. baumannii* 2021CK-01407 (CP104448.1) when BLASTed (Fig. 3A).

Between 135 and 250 kb on contig 1, methyltransferases, IS*Aba1* transposase and *gyrB* were found, without any other resistance gene (Fig. S5). A *bla*$_{OXA-51}$ carbapenemase gene, in close synteny with an N-acetyltransferase and a *trmA* methylase, was found within 2.04–2.06 Mbp on contig 1 without any MGE (Fig. 3B). Toward the end of contig 1 (2.375 Mbp-end), however, *dfrA1, sat2*, and *aadA1* were bracketed by an *IntI2* class 1 integron, IS*256*, Tn*7*-like (Tn*sE*-Tn*SD*) and a truncated transposase (Fig. 3C). A BLAST analysis of this region showed that it aligned with only chromosomes from both *A. baumannii* and other Enterobacterales species such as *Proteus mirabilis, Enterobacter hormaechei, Shigella sonnei, Citrobacter freundii/gillenii, Providencia rettgeri, Morganella morganii, Escherichia coli*, and *Moellerella wisconsensis*.

Figure 4 shows the resistance genes and their genetic environments on contig 2. Notably, the beginning (0–15 kb; Fig. 2A) and end (1.8 Mb-end; Fig. 2C) of this contig has the same resistance genes and MGEs but in opposite directions/orientations: *dfrA:sat2:aadA:::*IS*26:*Tn*7*-Ts*E:*Tn*7*-Ts*D::integrase/recombinase:*Tn*SA* endonuclease in the 5′–3′ direction and Tn*SA* endonuclease:integrase/recombinase::Tn*7*-Ts*D:*Tn*7*-Ts*E:*IS*26:::aadA:sat2:dfrA* in the 3′–5′ direction. Between these two repeated regions, ~714–716 kb, is the *bla*$_{ADC}$:IS*Aba1*resistance gene and IS (Fig. 4B). A BLAST

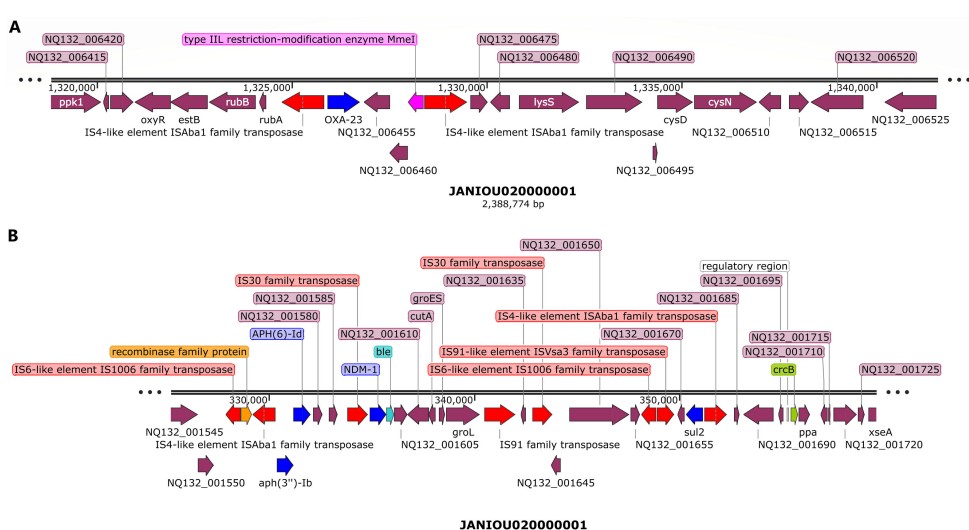

**FIG 2** Genetic environment of *sul2, aph(3″)Ib, aph(6)-Id*, and *blaOXA-23* and *blaNDM-1* carbapenemase on chromosomal contig 1 (~1.32–1.34 Mb, ~320–360 kb).*bla*$_{OXA-23}$ (shown as blue arrow) in panel A, is bracketed by a composite transposon consisting of IS*Aba1* and transposases. *sul2, aph(3″)Ib, aph(6)-Id, and bla*$_{NDM-1}$, in panel B (shown as red arrows), were sandwiched between IS*1006*, recombinase, IS*Aba1*, IS*30*, and IS*Vsa3* insertion sequences and transposases.

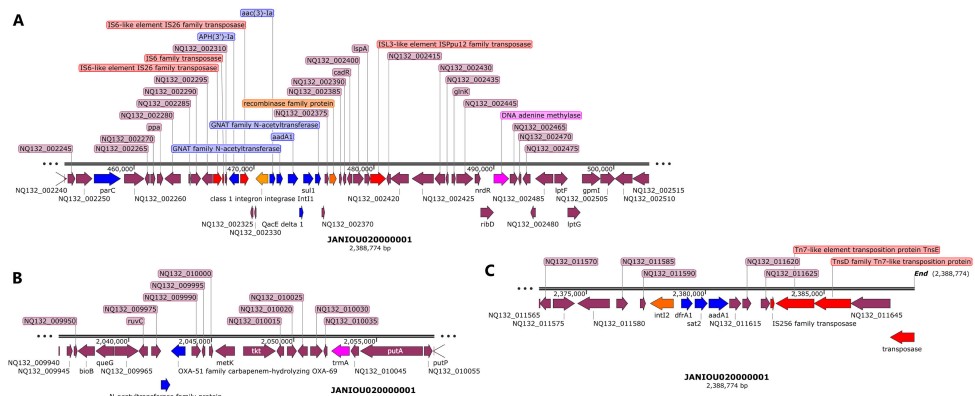

**FIG 3** Genetic environment of *aph(3')-Ia, aac(3)-Ia, aadA1, sat2, dfrA, and Sat2* resistance genes on chromosomal contig 1 (~450–500 kb, 2.04–2.06 Mb, ~2.375–2.39 Mb). IS*26*-IS*6* and IS*Ppu12* transposase bracketed *aph(3')-Ia, aac(3)-Ia,* and *aadA* while a recombinase and class 1 integron integrase (*IntI1*) sandwiched *sul1, qacE*, GNAT family N-acetyltransferase, and *aadA1* genes. Resistance genes are shown as blue arrows and mobile genetic elements are shown as red arrows. A DNA adenine methylase (purple arrow) was also found in close proximity to the resistance genes within the same genomic island on chromosome (A) N-acetyl transferase and *bla*<sub>OXA-51</sub> carbapenemase were also found in close proximity to *trmA* methylase (purple arrow) within the same region without any mobile genetic element (B) *dfrA, sat2,* and *aadA1* are also bracketed by a class 2 integron (*IntI2*), IS*256* and Tns*D/E* (Tn-*7*-like) transposases (C).

analysis of the two repeated regions shown in Fig. 2A through C showed that both regions aligned to chromosomes of *A. baumannii* and other Enterobacterales species.

R-B2.MM had two plasmids, contigs 3 (pR-B2.MM_C3) and 5 (pR-B2.MM_C6), identified through BLAST analyses of the contig sequences and found to be circular. While the plasmid type for pR-B2.MM_C6 was not identifiable, with only a *repM* replicase gene found on it, pR-B2.MM_C3 was found to contain IncFIB, IncX1, and IncFIC(FII) replicase gene sequences (Fig. 5; Fig. S6 and 7). Indeed, the pR-B2.MM_C6 plasmid had 100% coverage and nucleotide identity to *A. baumannii* plasmids such as CP145435.1 and CP142898.1 (Fig. S7) while pR-B2.MM_C3 plasmid had 65-77% coverage and 99% nucleotide identity with *E. coli* and *K. pneumoniae* plasmids (Fig. S6; Data set S1). The BLAST analyses of pR-B2.MM_C6 show that the plasmids it aligned 100% to were more than 17,462 bp long while pR-B2.MM_C6 was 8731 bp with 11 protein genes. pR-B2.MM_C3 was 193,714 bp with 184 protein-coding genes and only aligned with high nucleotide homology (99%) with sections of *E. coli* and *K. pneumoniae* plasmids (Fig. 5; Table S3).

*AadA1, bla*<sub>TEM-1B</sub>, *mcr-1, cmlA1, qnrS1, sul3, tet(M), tet(A),* and *dfrA* were found on pR-B2.MM_C3 while pR-B2.MM_C6 had no resistance gene. The resistance genes on pR-B2.MM_C3 were localized together in a genomic island (between 0 – 42 kb and 155–0 kb) (Fig. 5; Fig. S5 and 6) with *mcr-1* being flanked by IS*Apl1* and IS*903B* transposase. *tet*(M) and the other resistance genes on this plasmid were also flanked by a class-1 integron and recombinase, which were also bracketed by batteries of Tn*3* composite transposons and insertion sequences (ISs) (Fig. 5A and C). Within this genomic island were site-specific MTAses and restriction endonucleases (REs) (Fig. 5; Fig. S6). Although mobile genetic elements (MGEs) such as IS*1A*, IS*lB*, and IS*3* were also found within the 90 kb – 150 kb region, only a Mig-14 resistance gene was found in this region (Fig. 5B).

## Phylogenomic analysis

The strain was not significantly related to any of the antibiotic-resistant strains from Africa used in the phylogenomic analysis (Fig. 6). It was closely related to two strains (with a bootstrap value of 5): *A. baumannii* 13367 and 13259. Expectedly, these two strains had different STs from each other and R-B2.MM. There was also little uniformity in their resistomes, with *A. baumannii* 13367 having more similarity to R-B2.MM's resistome (Fig. 6). Among the global *A. baumannii* strains, however, there were significant

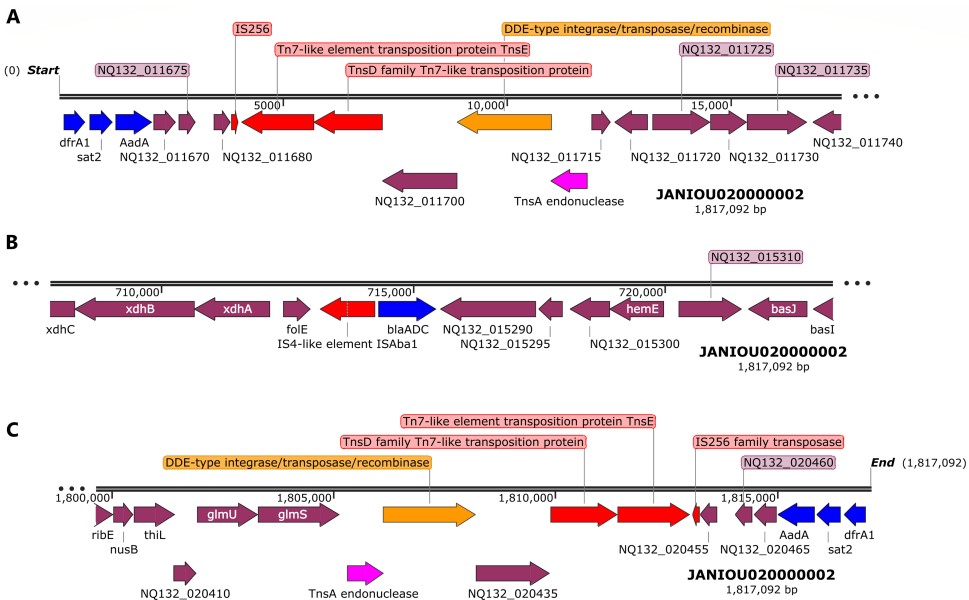

**FIG 4** Genetic environment of *aadA1, sat2, dfrA*, and *blaADC* resistance genes on chromosomal contig 2 (0–15 kb, 710–720 kb, and 1.8 Mb–1.82 Mb). IS256, Tn*sE*, and tns*D* mobile genetic elements (shown in red arrows), recombinase (orange arrow), and an endonuclease (purple arrow) bracketed *dfrA, sat2, aadA* (A) while *bla*$_{ADC}$ was in close synteny with IS*Aba1* (B) The region shown in panel C is a "cut and paste" transposition of the region shown in panel A but in the reverse orientation, suggesting that a replicative transposition event occurred within the genome.

evolutionary relationship between *A. baumannii* strains Ab905, Ab241, Ab238, A3232, and AbCTX19 (Fig. 7 ) as confirmed by the high bootstrap value of 100. *A. baumannii* Ab905 and Ab241 strains isolated from blood from Israel had the same resistome, but the remaining strains had similar but not the same resistome. Notably, *bla*$_{ADC}$, *bla*$_{OXA-71/69}$, *bla*$_{OXA-23}$, *sul1*, and *gyrA* S81L mutation were common among most of the strains. *A. baumannii* strains A3232 (Greece), AbCTX19 (France), and R-B2.MM had the same ST and belonged to the same clone.

*A. baumannii* strains that aligned closely with this study's strain, with more than 80% nucleotide homology and 70% query length with sections of R-B2.MM genome, were used to conduct a phylogenomic reconstruction analysis. As shown in (Fig. S9; Data set 2), 33 strains had a very close evolutionary relationship with this study's strain, most of which was of ST231 or ST1 clone. Although these strains were isolated from different sources, different years, and different countries, they were closely related to each other. However, the resistome was not conserved across these strains.

## Epigenomics

REBASE identified types I (M.Aca7364II and M.Aba0083I) and II (M.Aba858II) methyltransferases (MTAses), with types III and IV being absent: both types were found on only contig 1 (chromosome). A specificity subunit (S.Aba0083I) was also found on contig 1 (Data set 4). The recognition sequence of these REs was different except for only M.Aba0083I and S.Aba0083I. GATC motif was identified by PacBio with N6-methyladenosine (m6A) modifications (Data set 4). Type IIL restriction modification enzyme MmeI, homocysteine S-methyltransferase family protein (Fig. 1 and 2), Tr*mA* and DNA adenine methylase (Fig. 3), Tn*sA* endonuclease (Fig. 4), and site-specific DNA-methyltransferase and RE on pR-B2.MM_C3 contig 3 (Fig. 5) were annotated throughout the genome. The annotated MTAses and REs were mostly associated with the MGEs within the composite transposons or flanked by ISs.

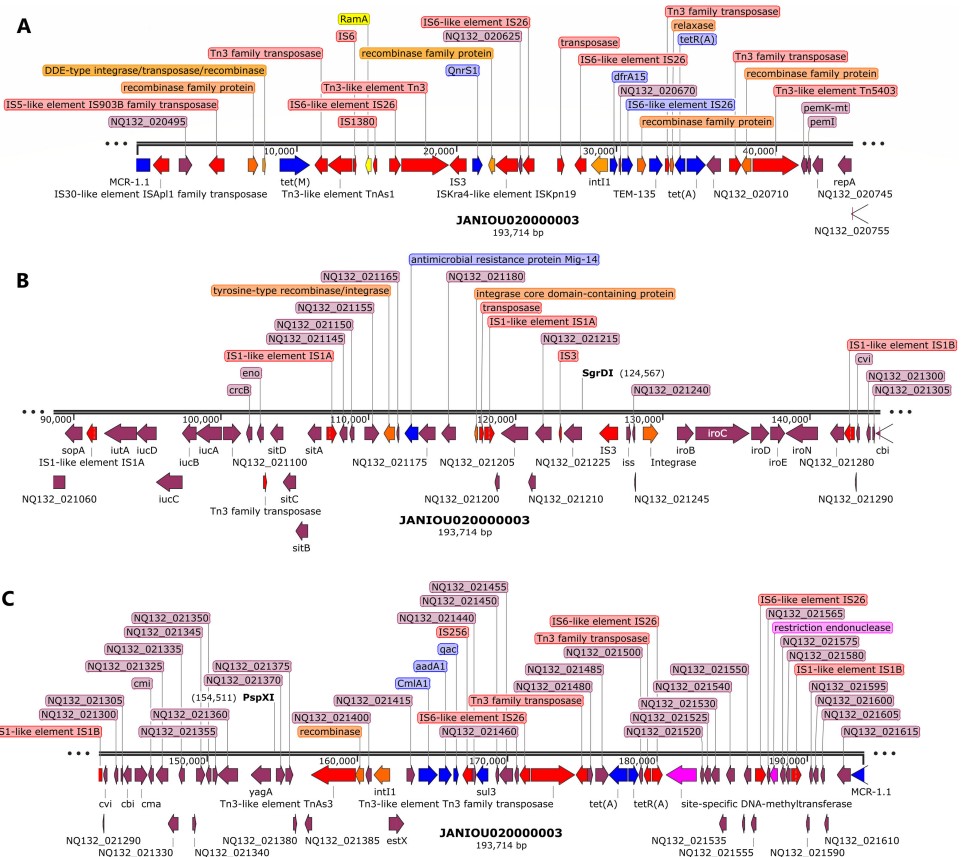

**FIG 5** Genetic environment of *mcr-1, tet(M), QnrS1, dfrA15, blaTEM-135, tet(A), cmlA1, aadA, qaC*, and *sul3* resistance genes on plasmid pR-B2.MM_C3 (contig 3). The resistance genes (blue arrows) were clustered together on a genomic island on pR-B2.MM_C3 at 0–42kb and 155 kb-0kb and sandwiched between composite transposons (red arrows) and integrons (orange arrows). The mobile genetic environment is comprised of composite transposons such as IS*903B*, IS*26*, IS*1380*, IS*1A*, *and* Tn*3*, alongside recombinases and class 1 *IntI1* integron. Site-specific DNA methylases were also found within this genomic island.

## Differentially expressed genes

The fold changes of each coding gene's expression levels are shown in supplementary data set 3. Out of the 4,220 coding genes, 261 were significantly expressed while 3,959 were not significantly expressed. As shown in the volcano plot in Fig. 8 and the summarized bar chart in Fig. S8, most of the significantly expressed genes were hypothetical proteins with unknown functions, followed by LysR family transcriptional regulators, phage replication proteins, class A beta-lactamases, GNAT resistance genes, outer membrane proteins, type I RMS, integrases, ABC/MFS efflux transporters, type-6 secretion systems (TSSS), OprD porins, etc. There were also MTAses, endonuclease III, MGEs, ABC/RND transporters, and prophages that were not significantly expressed (Data set 3). Notably, the following resistance genes were significantly highly expressed: $bla_{NDM}$, *QacE, sul2,* recombinase, aminoglycoside 3′-phosphotransferase, *aph(3′)-III/aph(3′)-IV/aph(3′)-VI/aph(3′)-VII*, aminoglycoside 3″-phosphotransferase, *aph(3″)-I*, macrolide 2′-phosphotransferase *mph(E)/mph(G)* family, and a class A beta-lactamase.

A genome map of the strain and annotations showing the GC content, genes/ORFs, resistance genes, phageome, horizontal gene transfer events and MGEs, and CRISPR/Cas systems mainly confirmed the findings above as well as showed the location of the various ARGs, MGEs, and horizontal gene transfer events on the genome. Notably, quinolone resistance-conferring mutations in *par*C and *gyr*A as well as efflux

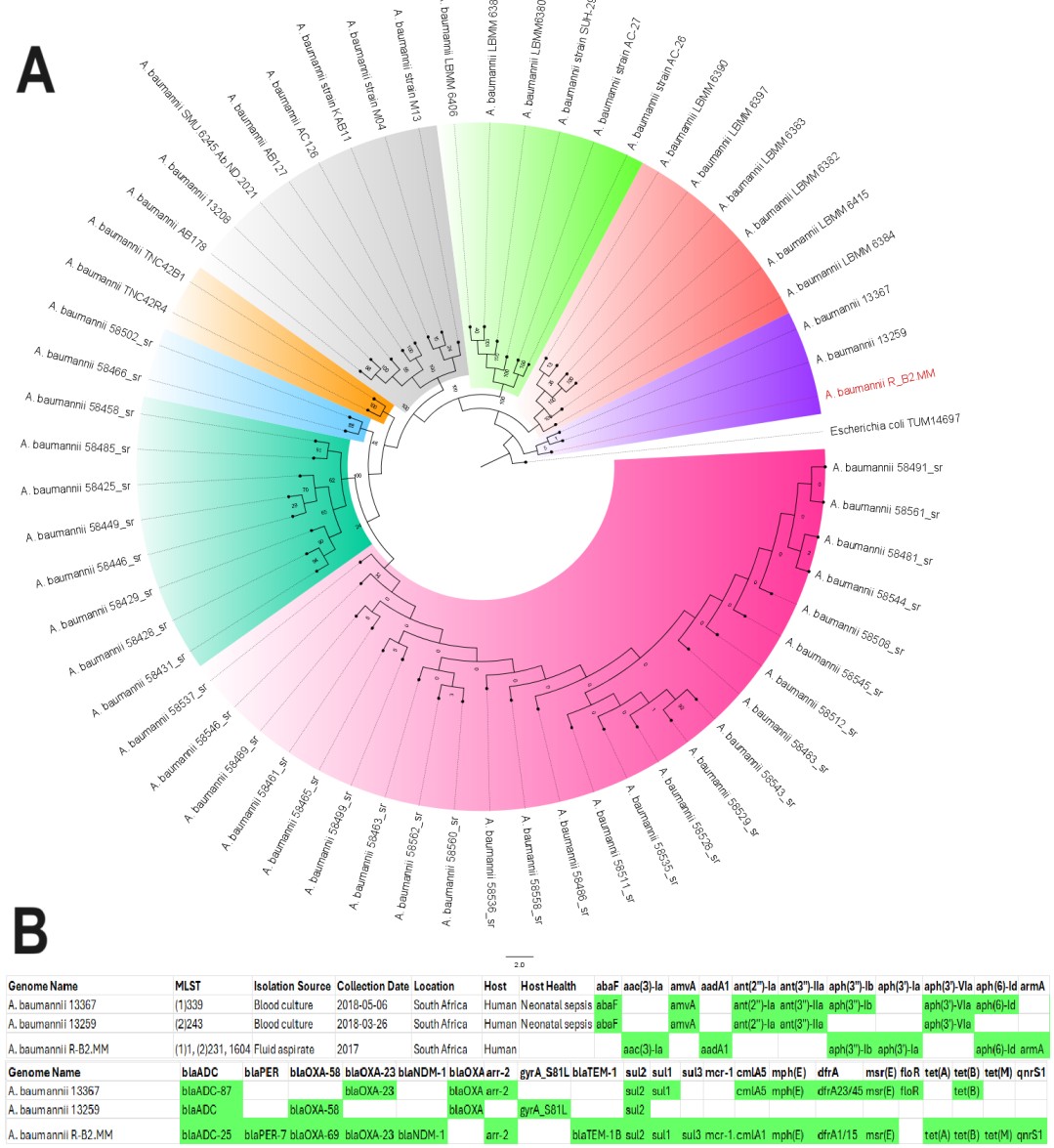

**FIG 6** Phylogenetic analysis of antibiotic-resistant *Acinetobacter baumannii* strains from Africa. The R-B2.MM strain was not closely related to any resistant strain in Africa. The most closely related strains (*A. baumannii* 13259 and 13367) were not supported by the bootstrap values to be significant. The R-B2.MM strain is shown as red while all other strains are shown as black. Bootstrap values of ≥50 is significant. The resistomes of the three strains are also shown in a table below the tree, with *A. baumannii* 13367 having more similarity with this study's strain. MLST (1) is the Pasteur Institute typing scheme while MLST (2) is the PubMLST typing scheme.

pumps mediating multidrug resistance were identified by the comprehensive antibiotic resistance database (CARD) and Prokka annotation (31, 32). A type IF CRISPR Cas gene was also present in the genome (Fig. S10 through S16.

## DISCUSSION

We report on the transcriptome, resistome, mobilome, epigenome, and evolutionary biology of a pandrug-resistant *A. baumannii* clinical strain that harbored $bla_{NDM}$, $bla_{OXA-23}$, $bla_{OXA-69}$, $bla_{ADC}$, $mcr-1$, and a plethora of resistance genes. To our knowledge, this is the first CRAB strain to co-harbor three carbapenemases, that is, $bla_{NDM}$, $bla_{OXA-23}$, and $bla_{OXA-69}$, and an $mcr-1$ globally (20, 33), with $mcr-1$ being found on a plasmid. Although $bla_{NDM}$ and $bla_{OXA-23}$ were not found on plasmids, they were found within

composite transposons within sections of the chromosome that aligned with plasmids, suggesting that they might have been integrated into the chromosomes from plasmids to form episomes. This suspicion is confirmed by the replicative transposition of the composite transposon genomic islands to other loci of the genome to form multiple genomic copies of the same genes or composite transposons (Fig. 1 through 4). It is further corroborated by the alignment of these genomic islands to Enterobacterales species such as *E. coli, K. pneumoniae, Serratia* sp., *Providencia* sp., etc. Indeed, *mcr-1* was also found within a composite transposon genomic island that can also be transposed from the plasmid to the chromosome (Fig. 5).

We are therefore of the opinion that this strain's genome underwent multiple replicative transposition events mediated by the composite transposons, recombinases, and integrons that blanketed the resistance genes, resulting in the duplication of resistance genes across the genome. The strain's chromosome also contains several plasmid-integrated regions or resistance genomic islands, forming episomes that also align to other Enterobacterales species and plasmids. The evolutionary biology and phylogenetic analysis of the strain, as well as the comparative resistome analysis with other closely related strains and clones, show that the resistome of our strain is quite unique (Fig. 6 and 7; Fig. S9). Hence, it is obvious that the genomic rearrangements observed in this genome are not wholly vertically transferred although it can vertically transfer this genome to other daughter cells as it multiplies. It can also easily transfer these resistance genomic islands horizontally to other cells through the multiple MGEs such as IS, transposons, recombinases, integrases, and plasmids found in the genome (Fig. S7; Fig. 1 through 5).

Instructively, there was no strain from South Africa and Africa, published to date, that was closely related to this study's strain while international strains were found to be within the same clade and of the same clone as this study's strain (Fig. 6 through 7; Fig. S9). This suggests that this strain might have been imported from abroad. Notably, the other international strains were found to be of the same clone and clade as R-B2.MM were isolated from different clinical sources such as blood, wound, urine, respiratory specimen, rectal swabs, and fluid aspirates, from different time periods spanning 1982 to 2023, and from different countries located across all continents: South and North America, Asia and the Middle East, Europe, and Australia. While this speaks to the wide geographical distribution of these strains and clade, and portends the worrying spread of MDR *A. baumannii* strains, their non-uniform resistome is also revealing. As discussed in the previous paragraph, the non-homogeneity of resistomes across the clade shows independent evolution of resistance traits as the strains spread across the globe.

The intra-genomic evolution seen in our strain, therefore, suggests that its exposure to antibiotics during treatment might have induced the replicative transposition events observed in the genome. This further supports the need to be measured in antibiotic administration to reign in the evolution and dissemination of antibiotic resistance within and across strains and species. Indeed, the transcriptomic data lend further evidence to this assertion in that we observed significant hyperexpression of genes involved in antibiotic resistance, including resistance genes, MGEs, MTAses, REs, transcription factors, membrane-associated proteins, T6SS, phage-associated genes, etc. (Data set 3). Evidently, the exposure to antibiotics does not only cause an increase in resistance gene expression but also in selected efflux pumps, regulatory genes, and MGEs that can both help the cell to survive by expelling xenobiotics and adapt its genome through accelerated transposition and transcription events to confer resistance.

This argument is supported by the presence of RMS (MTAses and REs) within the resistance genomic islands or in close synteny to the composite transposons. Given the important regulatory role of RMS in initiating or inhibiting transcription of key genes through DNA methylation, it is evident that its close association with the MGEs and its significant expression support its involvement in the observed transposition events and resistance. It has already been proven that exposure to low-dose antibiotics, which lead to antibiotic-resistance mutations and adaptations, is mediated epigenetically through

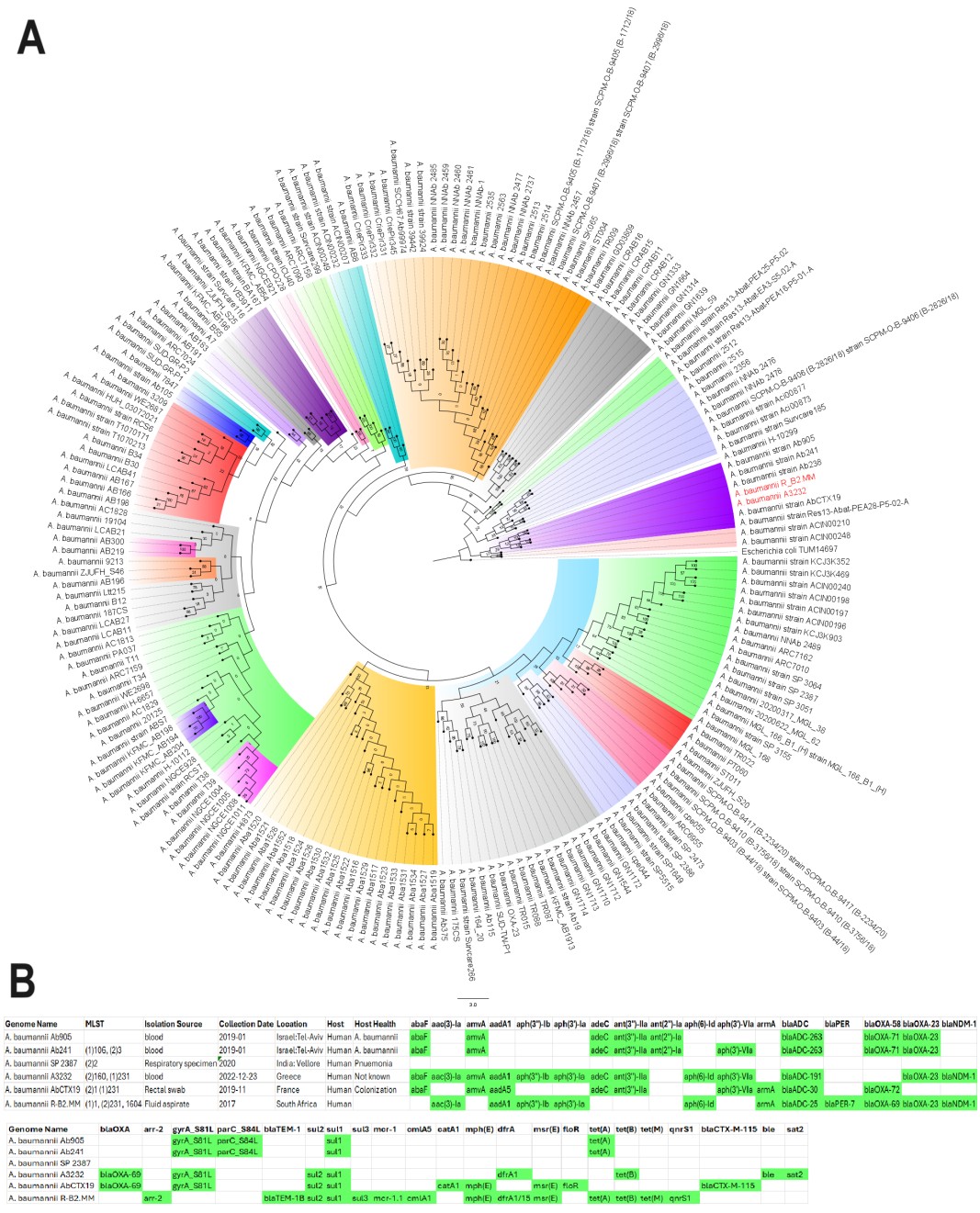

**FIG 7** Phylogenetic analysis of antibiotic-resistant global *Acinetobacter baumannii* strains. *A. baumannii* Ab905 (from Tel-Aviv, Israel, in 2019), Ab241 (from Tel-Aviv, Israel in 2019), Ab238 (from Tel-Aviv, Israel, in 2019), AbCTX19 (from Le Kremlin Bicetre, France, in 2019), and A3232 (Greece, 2022) strains formed the same clade with the R-52.MM strain. All these strains were isolated from humans and mostly from blood except strain AbCTX19 (rectal swab). Ab241 and Ab238 were of the same MLST (106 or 3) while AbCTX19 and A3232 were of MLST 231 (and 1 or 160, respectively). The significance of the clade was confirmed by the bootstrap. A3232 was most closely related to R-B2.MM evolutionarily and is thus shown as red on the tree. Bootstrap values of ≥50 is significant. MLST (1) is the Pasteur Institute typing scheme while MLST (2) is the PubMLST typing scheme.

the RMS (34, 35). Hence, the unique genomic arrangements and MGEs found in this isolate, which makes it different from other clones within the same clade, could most likely be mediated epigenetically. Adaptive resistance, which is mediated by epigenetic RMS factors, is transient in nature and tends to disappear in the absence of the triggering factor. This further supports our assertion that this strain independently developed its

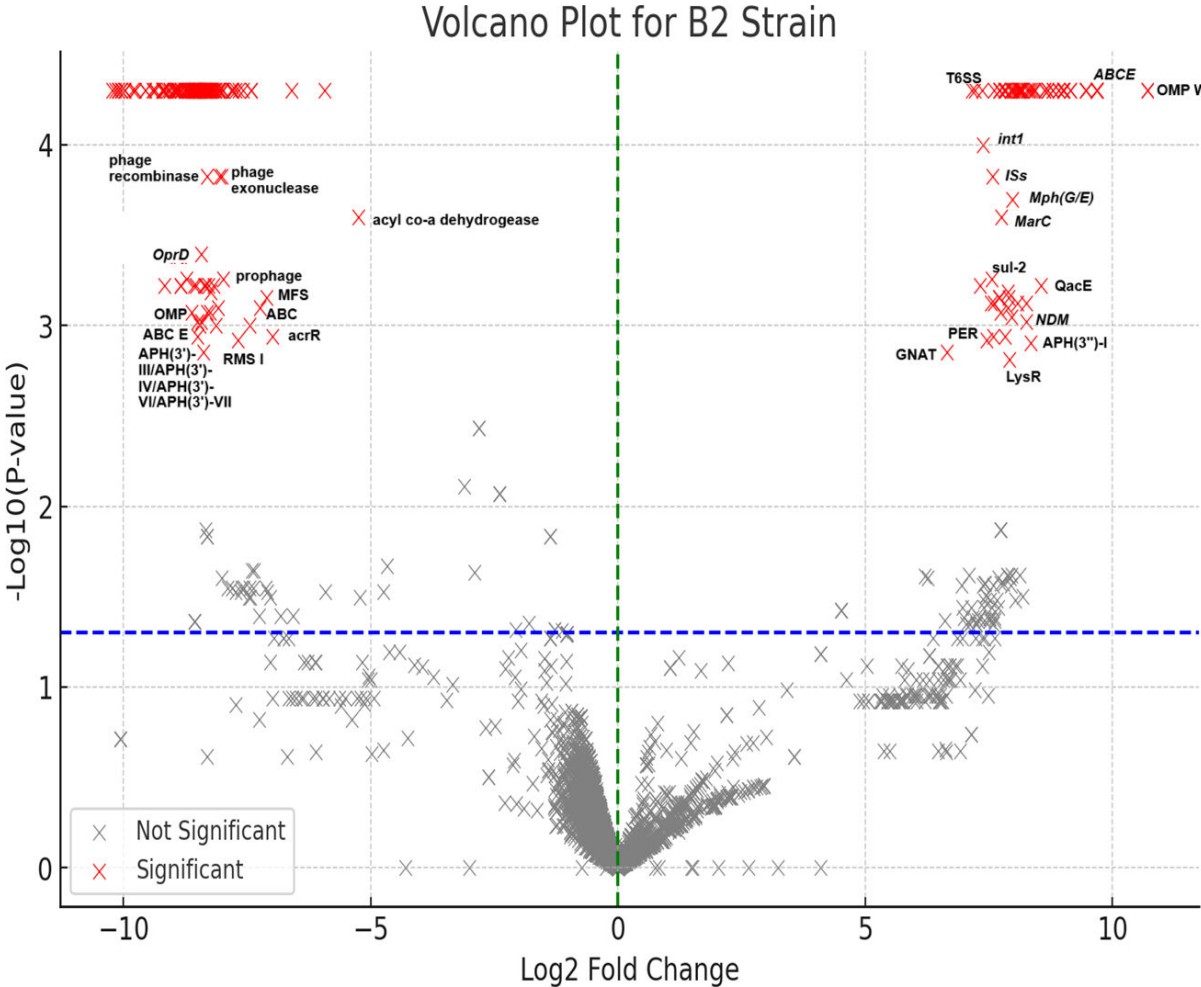

**FIG 8** A volcano plot showing differentially expressed genes (DEGs) in *Acinetobacter baumannii* R-B2.MM was exposed to colistin and carbapenems. The significant DEGs are shown in red while the non-significant DEGs are shown in gray. The significant DEGs include genes within the category of hypothetical protein, LysR transcriptional regulator, phage replication protein, aliphatic sulfonate monooxygenase, aldehyde dehydrogenase. Putative lipoprotein, urea carboxylase-related aminomethyltransferase, ribulose-5-phosphate 4-epimerase, and VgrG protein. The annotation on the plot shows highly expressed genes such as ABC efflux (E) OMP W (outer membrane protein W), *int1* (integrase/class 1 integron), resistance genes (*sul-2*, *bla*$_{NDM}$, *bla*$_{PER}$, GNAT acetyl transferase), ISs (insertion sequences, transposons), and type VI secretion systems (T6SS). Highly repressed genes included Type I RMS I, outer membrane porins (OprD) and proteins (OMP), prophage elements, acrR transcriptional regulators, and ABC and MFS efflux pumps.

MGE-mediated genomic evolution from antibiotic therapy using the epigenetic RMS pathways to regulate the transcription of key genes (34, 35).

Phenotypically, not all the EPIs resulted in a reduction in MICs of both ertapenem and colistin. Whereas EDTA and CCCP reduced colistin's MIC, EDTA alone could reduce the MIC of ertapenem. This is expected as NDM and MCR-1 are zinc-based metallo-β-lactamases, meaning that NDM and MCR-1 cannot function enzymatically without zinc (36).

Hence, the ability of EDTA to chelate zinc will prevent NDM and MCR-1 from being able to, respectively, hydrolyze their substrate antibiotics or transfer a phosphoethanola-mine (PEtN) residue to lipid A in the outer membrane (37, 38). The ability of CCCP, on the other hand, to inhibit colistin resistance is believed to be due to its ability to depolarize the plasma membrane and reduce ATP production (37). As colistin depolarizes the cell membrane, making it more permeable to leakages, CCCP seems to work in synergy with colistin. Hence, in the presence of CCCP, MCR-1 may be unable to counter the effect of colistin by adding PEtN to lipid A (37). Furthermore, the inability of the other EPIs to reduce ertapenem and colistin resistance was confirmed by the transcriptomic data in which many efflux pumps were not significantly expressed. Reserpine is also known

to mainly inhibit major facilitator superfamily (MFS)-type efflux pumps in Gram-positive bacteria (39). Hence, its inability to affect the MICs is expected. Verapamil and PAβN are, respectively, known to target MATE and RND pumps in Gram-negative bacteria (39). However, there was no significant expression of the MFS, MATE, and RND efflux pumps from the transcriptomic data, which explains why these EPIs did not have any effect on the MICs (Data set 3).

The repertoire of resistance genes found within the genomes correlated with the resistance phenome observed in the strain, showing that the resistance genes were being expressed to confer resistance to their respective antibiotic targets. This was also confirmed by the transcriptomic data with regards to the efflux, membrane protein, and resistance genes hyperexpression (Data set 3). Furthermore, the association of these resistance genes on gene cassettes or within the composite transposon shows that they are moved together with the MCR-1 and carbapenemases during the transposition events or horizontal gene transfer. Hence, it is expected that this pandrug-resistant strain can share its rich resistome with other commensals or pathogens within the same niche. This will become particularly so should such a population become exposed to antibiotics, which can trigger the epigenomic and transcriptional activity toward a resistant phenotype (34, 35). One major limitation of this study was our inability to undertake a plasmid conjugation experiment to determine the transferability of the plasmid found in this strain. Yet, the clone of R-B2.MM, ST1/ST231, is found worldwide, according to the PubMLST database, in humans with a single case, ST1/ST231, collected from the environment in Croatia. All three sequence types have been reported in Africa, specifically Ethiopia, Kenya, and Ghana and, only ST1 has been previously reported in South Africa (Pretoria) in 2010 (40).

The immediate genetic environment of the ARGs corroborated what has already been reported globally. For instance, *mcr-1* was flanked by IS*Apl1; tet*(A) and *tet*(M) by an IS*6*/IS*26*-Tn*3; sul* and *dfrA* by a class-1 integron-IS*26*; $bla_{NDM}$:*ble, aph(6´)-Id* and *aph(3´´)-Ib* by IS*Aba1*-IS*91*-IS*IS*30; $bla_{OXA-23}$ by IS-4-like IS*Aba1*; and *aadA1, qacE* and *sul1* by *Inti1*-IS*6* (30, 33, 41). The *mcr-1* was found on an IncF-IncX hybrid plasmid while most *mcr-1* genes are found on IncH and IncC (33, 42).

While the resistance genes identified in this strain have been reported globally, they are particularly common in specific areas and regions. Specifically, $bla_{NDM}$ is commonly reported from the Indian sub-continent and Southeast Asia where it first originated. There have also been substantial incidences in Europe, Africa, and North America (30, 43–45). The $bla_{OXA-23}$ and $bla_{OXA-48-like}$ resistance genes are also distributed globally but are particularly common in Europe, Turkey, Southeast Asia, and parts of Africa. In Europe, OXA-48-like producing Enterobacteriaceae (38%) are second to KPC-producers (42%), but more common than NDM-producers (12%) (33, 43, 46). *mcr* genes, however, have been identified in 72 countries in mostly *E. coli, K. pneumoniae, Salmonella* sp. in animals, humans, and the environment. They are specifically commonly reported in China and Europe (34, 43).

Owing to the clinical importance of carbapenems and colistin as last-resort antibiotics, these ARGs reduce the efficacy and spectrum of current treatments, increase treatment costs and mortalities, and lengthen hospital stays (47–49).

The absence of Type III and IV RMSs is confirmed by other studies (30, 33). Most of the REs and MTAses were found on the chromosomes (30) within the genomic islands or in very close synteny to the ARGs. However, not all the MTASes and REs were hyperexpressed significantly (Data set 4), suggesting that only a few of the epigenomic factors were triggered by the exposure to antibiotics. There were no orphan MTAses as REs were found on both the plasmid and chromosome alongside the MTAses within the genomic islands. This contrasts with other studies that found several orphan MTAses (without corresponding REs) (30, 50). No cytosine methyltransferase (Dcm) was found in the genome, meaning that cytosine is not methylated but adenine (Dam) is methylated at the GATC motif, resulting in a methylated adenine at the N6 position (m6A or $^{6m}$A). The GATC M6A motif is ubiquitous among prokaryotes (30, 33, 50)

DNA methylation in bacteria affects antibiotic resistance through several mechanisms. The first is through adaptive antibiotic resistance (AdR) where changes in DNA methylation patterns under stress conditions, including antibiotic exposure, can lead to transient and fast-appearing AdR phenotypes, such as overexpression of efflux pumps (35, 50–52). Second, the DNA mutation rate is related to DNA methylation as it is involved in DNA mismatch repair systems and methylated bases are mutational hotspots. Third, DNA methylation controls the expression of genes involved in antibiotic resistance. Specifically, Dam methylation is related to the expression of *acrD*, *marR*, *rpoS*, *fabI*, and *lrhA* genes in adaptive antibiotic-resistant strains. In *Vibrio cholerae*, the deletion of the orphan methyltransferase *vchM* is associated with groESL-2 upregulation and greater survival under aminoglycoside stress (52). These earlier findings show the importance of RMSs in the evolution of antibiotic resistance in susceptible strains during exposure to antimicrobial chemotherapy. This is very concerning as it demonstrates that ARGs are not the only defense employed by bacteria against antimicrobials.

Whereas some MTAses and REs were upregulated, some were not, suggesting that not all the RMSs were triggered by exposure to antibiotics. However, the functional summary of the differentially expressed genes (DEGs) shows that there are several unknown proteins that are marshaled in the face of antibiotic exposure to protect the bacterial cell from death. The significant DEGs with known functions included phage proteins and MGEs, outer membrane and efflux proteins, regulatory proteins and transcription factors, lipoproteins (useful for cell membrane structures), type-6 secretion systems, and ARGs. Instructively, 85 genes were upregulated while 176 were downregulated, resulting in 261 DEGs, indicating that a smaller fraction of the cellular machinery (6.2%) was marshaled to deal with the antibiotic threat. None of the RNAs in the genome was significantly expressed, which could be because colistin does not attack the RNA. Yet, it is intriguing that no RNA was hyper-expressed to produce more proteins.

## Conclusion

R-B2.MM encodes 35 resistance genes and twelve virulence genes, two plasmids (one of which is a hybrid IncX-IncF plasmid), and an MGE-rich chromosome with multiple resistance genomic islands that are episomal and aligns with plasmids and other Enterobacterales species. The strain was of closer evolutionary distance to several international strains suggesting that it was imported into South Africa. However, its resistome was unique, suggesting an independent evolution on exposure to antibiotic therapy mediated by epigenomic factors and MGE transposition events. The varied mechanisms available to this strain to overcome antibiotic resistance and spread to other areas and/or share its resistance determinants is worrying. This is ultimately a risk to public health as it was susceptible to only tigecycline. There is no better argument for antibiotic stewardship than the evidence provided herein to safeguard public health.

### ACKNOWLEDGMENTS

This work is based on the research supported wholly/in part by the National Research Foundation of South Africa under grant number: 131013. We are grateful to Dr. Busisiwe Lebogang Skosana for her assistance with the isolates' collection.

This work was funded by a grant from the National Health Laboratory Service (NHLS) given to Dr. John Osei Sekyere under grant number GRANT004 94809 (reference number PR2010486).

M.M. undertook laboratory work; N.M.M. was a co-supervisor to the study and assisted with funding; J.O.S. designed and supervised the study, undertook data and bioinformatic analyses and visualizations, and wrote and reviewed the manuscript.

### AUTHOR AFFILIATIONS

[1]Department of Medical Microbiology, School of Medicine, University of Pretoria, Pretoria, South Africa

²Department of Medical Microbiology, Tshwane Academic Division, National Health Laboratory Service, Pretoria, South Africa

³Institute of Biomarker Research and Clinical Development, Medical Diagnostic Laboratories, Genesis Biotechnology Group, Hamilton Township, New Jersey, USA

## AUTHOR ORCIDs

Masego Mmatli ⓘ http://orcid.org/0000-0002-5238-454X

John Osei Sekyere ⓘ http://orcid.org/0000-0002-9508-984X

## FUNDING

| Funder | Grant(s) | Author(s) |
|---|---|---|
| National Health Laboratory Service (NHLS) | GRANT004 94809 | John Osei Sekyere |
| National Health Laboratory Service (NHLS) | GRANT004 94808 | John Osei Sekyere |
| National Health Laboratory Service (NHLS) | GRANT004 94807 | John Osei Sekyere |

## AUTHOR CONTRIBUTIONS

Masego Mmatli, Data curation, Investigation, Methodology, Validation | John Osei Sekyere, Conceptualization, Data curation, Formal analysis, Funding acquisition, Investigation, Methodology, Project administration, Resources, Software, Supervision, Validation, Visualization, Writing – original draft, Writing – review and editing.

## DATA AVAILABILITY

This Whole-Genome Shotgun project, epigenomic, and RNAseq data have been deposited at DDBJ/ENA/GenBank under the BioProject number PRJNA861833 and accession number JANIOU000000000. The version described in this paper is version NZ_JANIOU020000000. The epigenomic and transcriptomic data can be found at GEO platform GPL19442 series GSE217148. The raw sequencing data can be also accessed at SRA number SRR22159804 (SRX18138690).

## ETHICS APPROVAL

This study was approved by the Research Ethics Committee of the Faculty of Health Sciences, University of Pretoria, with reference number 581/2020. Only stored clinical samples were used and no direct interactions with patients occurred. Written informed consent is taken by the hospital and diagnostic laboratory as part of the sample collection process to store and use the samples for research. The study was conducted according to the principles and protocols of the Declaration of Helsinki. All samples were deidentified to protect the identity and demographics of the patients.

## ADDITIONAL FILES

The following material is available online.

### Supplemental Material

**Data set S1 (mSystems01683-24-S0001.xlsx).** Tables S1 to S5.

**Data set S2 (mSystems01683-24-S0002.xlsx).** Metadata on demographics, clones (STs), biosamples, country of isolation, year and source of isolation, and resistance genes of strains from Africa and globally.

**Data set S3 (mSystems01683-24-S0003.xlsx).** Differentially expressed or repressed genes of *Acinetobacter baumannii* R-B2.MM.

**Data set S4 (mSystems01683-24-S0004.xlsx).** Methylation data, methylases, and DNA methylation motifs.

**Supplemental Figures (mSystems01683-24-S0005.pdf).** Figures S1 to S9.

Open Peer Review

**PEER REVIEW HISTORY (review-history.pdf).** An accounting of the reviewer comments and feedback.

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
