## [Reviewer comments · mSystems]

Plasmid-borne *mcr-1* and Replicative Transposition of Episomal and Chromosomal *bla*NDM-1, *bla*OXA-69, and *bla*OXA-23 Carbapenemases in a Clinical *Acinetobacter baumannii* Isolate

Masego Mmatli, Nontombi Mbelle, and John Osei Sekyere

Corresponding Author(s): John Osei Sekyere, University of Pretoria

Review Timeline:

Submission Date:	December 11, 2024
Editorial Decision:	January 1, 2025
Revision Received:	January 10, 2025
Accepted:	January 28, 2025

Editor: Karoline Faust

Reviewer(s): Disclosure of reviewer identity is with reference to reviewer comments included in decision letter(s). The following individuals involved in review of your submission have agreed to reveal their identity: Olga E. Khokhlova (Reviewer #1); Yingshun Zhou (Reviewer #2)

Transaction Report:

DOI: <https://doi.org/10.1128/msystems.01683-24>

Re: mSystems01683-24 (Plasmid-borne mcr-1 and Replicative Transposition of Episomal and Chromosomal blaNDM-1, blaOXA-69, and blaOXA-23 Carbapenemases in a Clinical Acinetobacter baumannii Isolate)

Dear Dr. John Osei Sekyere:

The reviewers now assessed the revised work and only minor comments remain to be addressed. To make the Volcano plot more informative, please also label selected significant genes in the plot.

Revision Guidelines

Sincerely,
Karoline Faust
Editor
mSystems

Reviewer #1 (Comments for the Author):

The main question addressed by the research: Mechanisms of antibiotic resistance and genetic structures of a clinical *Acinetobacter baumannii* isolate.

This topic « Plasmid-borne mcr-1 and Replicative Transposition of Episomal and Chromosomal blaNDM-1, blaOXA-69, and blaOXA-23 Carbapenemases in a Clinical *Acinetobacter baumannii* Isolate» relevant in the field of antibiotic resistant infection. This article summarized information about genetic environment, transcriptome, mobile, and resistome of multidrug-resistant clinical strain A. *baumannii*, isolated from patient in ICU.

The conclusion helps the reader evidence and arguments presented for understand the genomic evolution observed in strain A. baumannii, isolated from patient in ICU explains its adaptability and pandrug resistance and shows its genomic plasticity on exposure to antibiotics.

The references appropriate. The number of references (41) is enough, in addition, the number of sources five years ago (2019-2023) is 46,3% (19), which is enough.

The figures are informative and illustrative.

1. In this article authors ethical approval section has been added in lines 178-185. Conclusion of the ethical commission N^o 581/2020, but the strain isolated in 2017. What that mean?

Reviewer #2 (Comments for the Author):

1. The conjugation experiment can be used to observe conjugation frequency, but an additional component is needed.
2. The phrase "Types I and II methylases and restriction endonucleases were in close synteny to these resistance genes within the genomic islands" Can it affect the horizontal transmission of antibiotic resistance genes? Is there any evidence?
3. The sentence "Significantly expressed/repressed genes (6.2%) included resistance genes, hypothetical proteins, mobile elements, methyltransferases, transcription factors, membrane and efflux proteins." You need to focus more on which specific genes might be involved, rather than speaking in general terms.
4. The ANI between the isolate and the A.b ATCC1909. I don't quite understand why you chose ATCC1909 for your transcriptome analysis. If the genomic differences between the two strains are significant, their expression differences will be substantial, which could have a large impact on the results.

The main question addressed by the research: Mechanisms of antibiotic resistance and genetic structures of a clinical *Acinetobacter baumannii* isolate.

This topic « Plasmid-borne *mcr-1* and Replicative Transposition of Episomal and Chromosomal *bla*_{NDM-1}, *bla*_{OXA-69}, and *bla*_{OXA-23} Carbapenemases in a Clinical *Acinetobacter baumannii* Isolate» relevant in the field of antibiotic resistant infection.

This article summarized information about genetic environment, transcriptome, mobile, and resistome of multidrug-resistant clinical strain *A. baumannii*, isolated from patient in ICU.

The conclusion helps the reader evidence and arguments presented for understand the genomic evolution observed in strain *A. baumannii*, isolated from patient in ICU explains its adaptability and pandrug resistance and shows its genomic plasticity on exposure to antibiotics.

The references appropriate. The number of references (41) is enough, in addition, the number of sources five years ago (2019-2023) is 46,3% (19), which is enough.

The figures are informative and illustrative.

1. In this article authors ethical approval section has been added in lines 178-185. Conclusion of the ethical commission № 581/2020, but the strain isolated in 2017. What that mean?

Comments	Response
Editor	
The reviewers now assessed the revised work and only minor comments remain to be addressed. To make the Volcano plot more informative, please also label selected significant genes in the plot.	Thanks. I have annotated the volcano plot with a selection of the most re-pressed and expressed genes
Reviewer 1	
1. In this article authors ethical approval section has been added in lines 178-185. Conclusion of the ethical commission № 581/2020, but the strain isolated in 2017. What that mean?	The strain was isolated from the patient in 2017 and stored in our pathology laboratory as part of routine diagnostic procedures. However, the study into the resistance mechanisms was initiated in 2020, during which time the ethical approval was obtained.
Reviewer 2	
1.The conjugation experiment can be used to observe conjugation frequency, but an additional component is needed.	Thanks. We agree to the importance of the conjugation experiment. However, we are unable to perform it currently owing to constraints in resources
2. The phrase" Types I and II methylases and restriction endonucleases were in close synteny to these resistance genes within the genomic islands" Can it affect the horizontal transmission of antibiotic resistance genes? Is there any evidence?	We did not test for this and do not have evidence of the effect of the location of these RMS on HGT and ARGs is in the literature. We however found it important to point this (the closer location of the RMS to the ARGs) out owing to the effect of MTases on resistance genes expression. Being part of the immediate genetic environment of the ARGs, they were worthy of mention.
3. The sentence " Significantly expressed/repressed genes (6.2%) included resistance genes, hypothetical proteins, mobile elements, methyltransferases, transcription factors, membrane and efflux proteins. "You need to focus more on which specific genes might be involved, rather than speaking in general terms.	Thanks. I would gladly expand on this list as already done in lines 340 – 352 under "Differentially Expressed Genes". However, the Abstract does not allow more than 250 Words and doing so will exceed the Abstract word count.
4. The ANI between the isolate and the A.b ATCC1909. I don't quite understand why you chose ATCC1909 for your	The A. baumannii ATCC1909 was chosen because it is a susceptible and reference model strain (type species) for this

transcriptome analysis.

If the genomic differences between the two strains are significant, their expression differences will be substantial, which could have a large impact on the results.

particular species. As this study was mainly looking at the resistance determinants, we chose a strain that is known to be susceptible to almost all antibiotics so that we can compare our more resistant strain to it to obtain the differential gene expression.

We agree that it would have been better to use the same strain but the strain was already resistant (on isolation from the patient) and we do not have a susceptible variant of our strain.

Re: mSystems01683-24R1 (Plasmid-borne mcr-1 and Replicative Transposition of Episomal and Chromosomal blaNDM-1, blaOXA-69, and blaOXA-23 Carbapenemases in a Clinical Acinetobacter baumannii Isolate)

Dear Dr. John Osei Sekyere:

I am pleased to inform you that your manuscript has been accepted, and I am forwarding it to the ASM production staff for publication. Your paper will first be checked to make sure all elements meet the technical requirements. ASM staff will contact you if anything needs to be revised before copyediting and production can begin. Otherwise, you will be notified when your proofs are ready to be viewed.

Sincerely,

Karoline Faust
Editor
mSystems